# Anti-Inflammatory Effect of Caffeine on Muscle under Lipopolysaccharide-Induced Inflammation

**DOI:** 10.3390/antiox12030554

**Published:** 2023-02-23

**Authors:** Tuany Eichwald, Alexandre Francisco Solano, Jennyffer Souza, Taís Browne de Miranda, Liebert Bernardes Carvalho, Paula Lemes dos Santos Sanna, Rodrigo A. Foganholi da Silva, Alexandra Latini

**Affiliations:** 1Laboratory of Bioenergetics and Oxidative Stress—LABOX, Department of Biochemistry, Federal University of Santa Catarina, Florianópolis 88037-100, Brazil; 2Epigenetic Study Center and Gene Regulation—CEEpiRG, Program in Environmental and Experimental Pathology, Paulista University—UNIP, São Paulo 05508-070, Brazil; 3School of Dentistry, University of Taubaté, Taubaté 12020-3400, Brazil

**Keywords:** cytokines, inflammasome, epigenetics, DNA methylation, adenosine receptors, bioenergetics

## Abstract

Evidence has shown that caffeine administration reduces pro-inflammatory biomarkers, delaying fatigue and improving endurance performance. This study examined the effects of caffeine administration on the expression of inflammatory-, adenosine receptor- (the targets of caffeine), epigenetic-, and oxidative metabolism-linked genes in the *vastus lateralis* muscle of mice submitted to lipopolysaccharide (LPS)-induced inflammation. We showed that caffeine pre-treatment before LPS administration reduced the expression of *Il1b*, *Il6*, and *Tnfa*, and increased *Il10* and *Il13*. The negative modulation of the inflammatory response induced by caffeine involved the reduction of inflammasome components, *Asc* and *Casp1*, promoting an anti-inflammatory scenario. Caffeine treatment *per se* promoted the upregulation of adenosinergic receptors, *Adora1* and *Adora2A*, an effect that was counterbalanced by LPS. Moreover, there was observed a marked *Adora2A* promoter hypermethylation, which could represent a compensatory response towards the increased *Adora2A* expression. Though caffeine administration did not alter DNA methylation patterns, the expression of DNA demethylating enzymes, *Tet1* and *Tet2*, was increased in mice receiving Caffeine+LPS, when compared with the basal condition. Finally, caffeine administration attenuated the LPS-induced catabolic state, by rescuing basal levels of *Ampk* expression. Altogether, the anti-inflammatory effects of caffeine in the muscle can be mediated by modifications on the epigenetic landscape.

## 1. Introduction

Regular, moderate-intensity exercise has been proven to promote an anti-inflammatory state that helps prevent the development of chronic diseases (for a review see [1]). Strenuous exercise can lead to increased levels of blood proinflammatory cytokines, which are linked to fatigue, and therefore, to reduced performance [2]. This scenario has led to caffeine administration being used to increase alertness [3], to accelerate metabolism [4], and to delay fatigue development in aerobic and anaerobic exercises, including muscular strength [5], running [6], cycling [7], and team sports [8], among others.

While it is unclear what molecular mechanisms are behind caffeine consumption and its ergogenic responses, evidence is mounting that caffeine may induce anti-inflammatory effects in both humans, and animals. For example, it has been demonstrated that caffeine supplementation reduced inflammatory markers in the blood of athletes [9,10,11]. In the case of animal models, reduced pro-inflammatory and increased anti-inflammatory markers were not only seen in the blood of trained rats, but also in key tissues linked to exercise performance, such as the brain, the lung, the heart and the skeleton, of rodents exposed to caffeine [12,13,14,15,16]. Furthermore, an elegant study involving 114 participants showed that caffeine intake is associated with lower inflammation and activation of the inflammasome, which resulted in less production of the pro-inflammatory cytokine interleukin-1 beta (IL-1b) [17].

In addition, caffeine supplementation has been shown to cause changes in gene expression that could be linked to improved exercise performance [18,19,20]. These modifications have been related to altered epigenetics, a term conceived to describe the possible causal processes acting on genes that regulate phenotype [21]. Some of the reported effects of caffeine are associated with DNA methylation [22], a major epigenetic factor influencing gene activities. Considering that epigenetics can change the activity of a DNA segment without changing the sequence, it is plausible that caffeine can modulate inflammatory processes by changing the epigenetic landscape. When DNA methylation is increased in a gene promoter, it will typically act to repress gene transcription, including the expression of inflammatory mediators. Altogether, we aimed to understand whether caffeine can modulate epigenetics to induce an anti-inflammatory scenario in the mouse skeletal muscle.

## 2. Materials and Methods

### 2.1. Animals

Adult Swiss male mice (3–5 months of age; body mass 45–50 g) from the central animal house of the Centre for Biological Sciences, *Universidade Federal de Santa Catarina* (Brazil) were kept in a controlled environment (22 ± 1 °C, 12 h light/dark cycle) with water and food ad libitum, for ten days (acclimatation period). The experimental protocols were approved by the Ethics Committee for Animal Research (PP00760/CEUA) of the Federal University of Santa Catarina (Brazil). All efforts were made to minimize the number of animals used and their suffering. Five mice were included per experimental group, unless otherwise stated.

### 2.2. LPS-Induced Inflammation

Acclimatized mice were randomly divided into the following 4 groups (5 animals per group): Vehicle: Animals that received an intraperitoneal (i.p.) injection of 0.9 % sodium chloride (injection volume of 0.1 mL for every 10 g of body weight); Caffeine: Animals that received an i.p. injection of caffeine (6 mg/kg of body weight); LPS: Animals that received an i.p. injection of LPS (0.33 mg/kg of body mass; *E. coli* LPS, serotype 0127:B8), and Caffeine+LPS: Animals that received an i.p. injection of caffeine and 15 min later received the injection of LPS. Twenty-four hours after the treatment mice were euthanized by cervical dislocation and the vastus lateralis muscle was immediately collected and processed in Trizol as previously published by our group [23]. The dosage of LPS used was based on previously published data [24,25].

### 2.3. RNA Extraction and cDNA Synthesis

For total RNA extraction, the *vastus lateralis* muscle was collected 24 h after the administration of the different treatments (vehicle, caffeine, LPS and Caffeine+LPS), and immediately homogenized with Ambion TRIzol Reagent (Life Sciences, Fisher Scientific Inc., Waltham, MA, USA). After adding 200 μL of chloroform, and followed by centrifugation at 17,000× *g* for 15 min at 4 °C, the upper aqueous layer containing the RNA was collected and transferred to a new tube. Then, 800 μL of chilled isopropanol were added and after light agitation, the RNA was precipitated by centrifugation at 17,000× *g* for 15 min at 4 °C. The supernatant was removed by inversion and the precipitated RNA was washed with 1 mL of 70 % alcohol and again centrifuged at 17,000× *g* for 5 min at 4 °C. Supernatants were discarded and 50 μL of nuclease-free H_2_O was added to the tube. The quantity and purity of the extracted RNA was estimated by using the NanoDrop spectrophotometer, at 260 and 280 nm. The synthesis of the cDNA was performed after treating the total RNA with DNase I (Invitrogen, Carlsbad, CA, USA), and with high-capacity cDNA Reverse Transcription Kit (Applied Biosystems, Foster City, CA, USA), according to the manufacturer’s instructions.

### 2.4. Real-Time Reverse Transcription and Quantitative PCR (RT-qPCR)

Real-time reverse transcription and quantitative PCR (RT-qPCR) reactions were performed using SYBR Green Master Mix (PowerUp™ SYBR™ Green Master Mix-Applied Biosystems, Foster City, CA, USA) with specific primers shown in Table 1. All the reactions were carried out in a total of 10 μL, containing 5 μL of specific primers (0.4 μM of each primer), 50 ng of cDNA and nuclease-free H_2_O in a QuantStudio^®^ 3 Real-Time PCR (Thermo Fisher Scientific, Waltham, MA, USA).

### 2.5. DNA Extraction

Genomic DNA (gDNA) was extracted from the mouse muscle 24 h after the treatments. Tissues were homogenized in extraction buffer (10 mM Tris pH 3.0; 0.5% SDS, 5 mM EDTA) and then digested with proteinase K solution at 65 °C for 16 h. Additionally, 500 μL of equilibrium phenol was transferred to the tube and thus the mixture was spun down at 17,000× *g* for 15 min. The upper aqueous layer containing the target DNA was preserved and mixed with 200 μL of chloroform. The mixture was centrifuged at 17,000× *g* for 15 min and the supernatant was collected and transferred to a new tube. Then, 800 μL of isopropanol and 150 μL sodium acetate 3 M was added to the mixture. Next, the mixture was centrifuged at 17,000× *g* for 15 min. The supernatant was removed, and the pellet was washed with 500 μL of 70 % alcohol, centrifuged at 17,000× *g* for 5 min. The supernatant was then completely discarded and 50 μL of nuclease free H_2_O was added to the tube. The quantity and purity of extracted gDNA was estimated by using the spectrophotometer apparatus NanoDrop, at 260 and 280 nm.

### 2.6. Enzymatic gDNA Treatment

After confirming the quantity and purity by spectrophotometry (OD 260/280 ≥ 1.8 and OD 260/230 ≥ 1.0), the gDNA was treated with T4-β-glucosyltransferase (T4-BGT) and subsequently with MspI and HpaII (New England BioLabs, Beverly, MA, USA). For this, three tubes (A, B and C) containing 400 ng gDNA of each sample were treated with 40 mM UDP glucose and T4-BGT (1 unit) for 1 h at 37 °C, followed by enzyme inactivation for 10 min at 65 °C. Next, the samples were digested with H_2_O (tube A), MspI (tube B) and HpaII (tube C) for 2 h at 37 °C according to the manufacturer’s instructions.

### 2.7. Methylation-Specific qPCR (MS-qPCR)

MS-qPCR methylation data were derived from 5 independent animals and a technical duplicate. The pattern of methylation (5-meC) and hydroxymethylation (5-hmeC) of the promoter regions of Adora1 (island 1 (F: 5′ AAG GAG CTC ACC ATC CTG 3′); (R: 5′ GTG GGT GGG CAC AGG GTA G 3′) and island 2 (F: 5′ CGA GAC TCC ACT CTG GC 3′); (R: 5′ CAC CTC GGT ACT GTC CCT GT 3′)) and Adora2A (F: 5′ AGG GTG CGC CCA TGA GCG GC 3′); (R: 5′ CAA CCC GAG AGT CTG ACC CGC CT 3′) were determined in qPCR reactions containing 2x SYBR Green I Master (5 µL), 0.4 µM specific primers (1 µL), 25 ng of treated gDNA (1.5 µL-3 conditions: H_2_O, MspI and HpaII) and q.s.p of nuclease-free H_2_O (2.5 µL). Primer sequences were designed in regulatory regions with CpG islands within regions of hypersensitivity to DnaseI, regulated by histone modification markers and with transcription factor binding sites using the Primer3 Input program (version 0.4.0) [26]. All primers sequences were blasted to confirm chromosomal location by the in-silico PCR tool (https://genome.ucsc.edu/, accessed on 15 June 2022) and the secondary structures and annealing temperatures analyzed using the Beacon Designer program (http://www.premierbiosoft.com/, accessed on 15 June 2022).

### 2.8. Statistical Analysis

Data are presented as mean ± standard error of mean (SEM). Data were analyzed by two-way ANOVA followed by the post hoc test of Tukey when *F* was significant. When comparing two independent groups, Student’s *t*-test for independent samples was used. The accepted level of significance for the tests was *p* < 0.05. Statistics and all graphs were performed by using GraphPad Prism 9^®^.

## 3. Results

### 3.1. Caffeine Administration Reduced LPS-Mediated Inflammation in the Mouse Muscle

Figure 1 shows the effect of caffeine and/or LPS administration (i.p.) after twenty-four h on pro-inflammatory cytokines gene expression in the mouse *vastus lateralis* muscle (Figure 1A). LPS exposure significantly increased the expression of the pro-inflammatory cytokine *Il1b* (*F*_(1,16)_ = 6.46, *p* < 0.01) (Figure 1B). Moreover, the expression of the anti-inflammatory cytokines *Il10* (*F*_(1,16)_ = 6.73, *p* < 0.05) (Figure 1E) and *Il13* (*F*_(1,16)_ = 5.36, *p* < 0.01) (Figure 1F) were also upregulated, possibly as a physiological compensatory response elicited by LPS-induced inflammation. Figure 1B shows that the expressions of *Il1b* and *Il6* (Figure 1C) were downregulated, and *Il10* (Figure 1E) upregulated when caffeine was administered in association with LPS. Furthermore, caffeine *per se* positively modulated the expression of *Tnfa* (*F*_(1,16)_ = 19.32, *p* < 0.001) (Figure 1D) and decreased the *Il*6 levels (*F*_(1,16)_ = 2.20, *p* < 0.05) (Figure 1C). However, no differences were observed in the levels of *Il1b* (Figure 1B), *Il10* (Figure 1E), and *Il13* (Figure 1F).

### 3.2. The Anti-Inflammatory Effect of Caffeine Was Mediated by Downregulating Nrlp3 Inflammasome Components

Figure 2 shows the effect of the administration of caffeine and/or LPS (i.p.) on NLRP3 inflammasome components gene expression in the mouse *vastus lateralis* muscle (Figure 2A). Figure 2 shows that LPS administration elicited the upregulation of the inflammasome assembly components *Asc* (*F*_(1,15)_ = 28.90, *p* < 0.001) (Figure 2C) and *Casp1* (*F*_(1,15)_ = 58.57, *p* < 0.001) (Figure 2D) in the mouse muscle, which was prevented by the administration of caffeine (Caffeine+LPS experimental group). Although, LPS treatment *per se* did not alter the levels of *Nlrp3* expression 24 h after the administration, the combination with caffeine provoked its upregulation (*F*_(1,16)_ = 34.78, *p* < 0.001) (Figure 2B) in the mouse muscle. No differences were induced by caffeine administration alone. Similar results were found in the absolute gene expression (Appendix A).

### 3.3. Caffeine Administration Enhanced the Expression of Adenosinergic Receptors in the Vastus Lateralis Muscle Mice

Figure 3 shows the effect of caffeine administration on adenosinergic receptors gene expression and gene methylation. Caffeine administration *per se* increased the expression of *Adora1* (*F*_(1,16)_ = 10.03, *p* < 0.05) (Figure 3A) and *Adora2A* (*F*_(1,16)_ = 16.26, *p* < 0.001) (Figure 3B) in the mouse muscle, which was prevented by the administration of LPS alone and combined with caffeine. LPS per se increased the levels of *Adora1* (*F*_(1,16)_ = 10.03, *p* < 0.01) (Figure 3B). Considering that gene expression can be controlled by epigenetics, the levels of DNA methylation and demethylation of *Adora1* and *Adora2A* were analyzed. While global DNA methylation was not modified by caffeine and/or LPS administration (Figure 3C,D), 5-meC/5-hmeC ratio, used as an index of DNA methylation, was increased under caffeine and Caffeine+LPS administration (*F*_(1,16)_ = 0.16, *p* < 0.05) (Figure 3E). The higher methylation of the *Adora2A* promoter might represent a homeostatic mechanism to control the upregulation of the gene. Appendix A shows that the absolute gene expression of the two ARs, *Adora1* and *Adora2A*, were similar in basal conditions.

### 3.4. Caffeine+LPS Exposure Enhanced the De Novo DNA Methylation in the Vastus Lateralis Muscle Mice

Figure 4 shows the effects of caffeine and/or LPS administration on the epigenetic profile in the mouse *vastus lateralis* muscle. Caffeine+LPS significantly upregulated the expression of the maintenance methylation gene *Dnmt1* (*F*_(1,16)_ = 101.05, *p* < 0.001) (Figure 4A), and de novo methylation gene *Dnmt3A* (*F*_(1,15)_ = 1.26, *p* < 0.001) (Figure 4B), while the expression of de novo methylation gene *Dnmt3B* was significantly downregulated (*F*_(1,16)_ = 0.06, *p* < 0.001) (Figure 4C) in the mouse muscle, when compared with the Vehicle and LPS groups. Caffeine+LPS treatment also inhibited the expression of the gene encoding the DNA demethylation enzyme *Tet3* (*F*_(1,16)_ = 133.5, *p* < 0.001) (Figure 4F), while no effect was observed on *Tet1* and *Tet2* when compared with the basal condition (vehicle), or under inflammation (LPS group). In addition, LPS modulated the expression of *Tet* genes; while LPS positively modulated the expression of *Tet2* (*F*_(1,16)_ = 11.84, *p* < 0.05) (Figure 4E), LPS administration negatively modulated the expression of *Tet3* (*F*_(1,16)_ = 133.50, *p* < 0.001), when compared with the vehicle condition. LPS treatment also compromised the expression of *Dnmt3B*, which was partially attenuated by caffeine co-administration (*F*_(1,16)_ = 0.06, *p* < 0.01) (Figure 4C). Caffeine treatment *per se* increased the levels of *Tet2* (Figure 4E), while it inhibited *Tet3* expression (Figure 4F).

### 3.5. Caffeine Administration Attenuated the Catabolic State Induced by LPS Administration in the Mouse VASTUS lateralis

Figure 5 shows the effects of caffeine and/or LPS catabolism in the mouse muscle. LPS treatment significantly elicited the upregulation of the energy status sensor gene *Ampk* (*F*_(1,16)_ = 5.35, *p* < 0.01) (Figure 5). The coadministration of caffeine significantly reverted the effect induced by LPS.

## 4. Discussion

Caffeine is a stimulant drug widely known and used due to its psychoactive and ergogenic effects [27]. The effects of caffeine on physical exercise, endurance performance, and fatigue stalling are well documented [28,29,30]. However, the molecular mechanisms behind these modulations are still under study. To the best of our knowledge, this is the first study to show that epigenetics is involved in the anti-inflammatory effects of caffeine on the *vastus lateralis* muscle of resting mice. Here, we showed that treatment with caffeine prevented an increase of the gene expression of LPS-induced pro-inflammatory cytokines *Il1b* and *Il6* and promoted the upregulation of the anti-inflammatory genes *Il10* and *Il13* in the mouse muscle. The anti-inflammatory state observed in the caffeine experimental group occurred with decreased gene expression of the NLRP3 inflammasome components, *Asc* and *Casp1*. Indeed, the activation of caspase 1 mediates the cleavage of pro-IL-1β to generate and release its biologically pro-inflammatory active form, IL-1β [31]. Moreover, caffeine administration promoted the upregulation of the adenosinergic receptors *Adora1* and *Adora2A*, the signaling of which is known to induce vasodilatation, healing and anti-inflammation, promotion of tissue blood flow and cellular homeostasis in different cell types [32]. Thus, in order to maintain homeostasis, the upregulation of *Adora2A* might have been responsible of triggering the methylation of its promoter. While DNA methylation patterns were not altered by caffeine treatment, the DNA methylating status was increased after Caffeine+LPS administration, suggesting that the observed adaptation to inflammation induced by caffeine was due to epigenetics.

Caffeine is the most commonly consumed social drug to increase alertness, arousal and energy [33]. Its consumption has been related to improvement in cognitive performance and mood in healthy population [34,35], and is the main ergogenic resource used by athletes to enhance exercise performance, extend time to exhaustion, and to delay fatigue [36].

Moreover, it has been shown that caffeine upregulated dopamine metabolism and signaling, and increased the synthesis and turnover of noradrenaline, being closely associated with an improvement in both peripheral and central fatigue [37]. Furthermore, the ergogenic effects of caffeine consumption have also been shown to be more evident in fatigued than in well-rested subjects [38,39]. These effects are proposed to be mediated by the non-selective antagonism of Adora1 and Adora2A [33]. Adora1 are widely expressed in the cortex, hippocampus, cerebellum, and thalamus [33], and in the adipose tissue, stomach, kidney, and heart [40]. Due to the capacity for lowering cAMP intracellular levels, the activation of Adora1 promotes bradycardia, inhibition of lipolysis, antinociception, reduction of sympathetic and parasympathetic activity, neuronal hyperpolarization, among others [41]. In contrast, Adora2A has a more restricted distribution, being more expressed in the striatum, nucleus *accumbens*, and olfactory tubercle [33]. Skeletal muscle, bladder, and the immune system are the tissues with the highest density of these receptors in the periphery [42]. The activation of Adora2A triggers neurotransmitter release, anti-inflammatory immune responses, and vascular smooth muscle cell relaxation, due to the activation of signaling pathways mediated by increased cAMP intracellular levels [36,41]. Therefore, the typical effects of adenosine, the natural agonist of Adoras and the final catabolite of ATP, that are associated with tiredness and drowsiness are counterbalanced by caffeine.

The effective dose of caffeine to antagonize Adoras and to lead to increased exercise time to fatigue ranges from 3 to 9 mg/kg in humans [28,43,44] and rodents [45,46]. These doses have been shown to increase performance in endurance, intermittent and resistance exercises in humans [5,6,7,8,47,48,49,50].

These effects have been associated with enhanced peripheral energy metabolism, activation of ryanodine channels for quicker release of calcium, and oxidant system in the muscle, improving muscle speed and strength (for a review see [16]). However, a large body of evidence suggests that caffeine can also mediate its ergogenic effects by inducing an anti-inflammatory status, preventing excessive endogenous catabolism and oxidative stress [1,16,51]. The anti-inflammatory effect of caffeine is also supported by the fact that individuals who suffer from cancer, obesity or liver, metabolic or neurodegenerative disorders and for whom persistent inflammation has been reported, developed fatigue when the symptoms appeared [52,53,54,55,56,57,58]. Moreover, high circulating levels of caffeine have been associated with delayed onset or reduced risk of dementia in individuals with mild cognitive impairment [59]. Furthermore, healthy individuals receiving an acute dose of caffeine showed reduced levels of pro-inflammatory markers and delayed development of fatigue [54,60]. Indeed, one of the first and most common symptoms associated with system immune activation is fatigue [61]. In addition, regular coffee consumption has also been associated with a reduced risk of low-grade inflammation in clinical conditions such as type 2 diabetes *mellitus* [62], and metabolic syndrome [63].

Considering that fatigue is characterized by temporary reductions in voluntary muscular force production, and cognitive and motivational changes that induces poor physical performance [64], we aimed to study whether caffeine could induce anti-inflammatory effects in the inflamed mouse muscle, and whether these effects are associate with epigenetic modifications.

It has been suggested that the development of fatigue may activate pathways that promote the activation of nuclear factor kappa b (NF-κB), which is considered a prototypical pro-inflammatory signaling pathway (for a review see [65]). NF-κB is known to be activated by a wide array of mediators, including LPS, inflammatory cytokines such as IL-1b and TNF-a, and reactive oxygen species, which in turn activates several signal transduction cascades and induces changes in transcription factors that promote a pro-inflammatory status. NF-κB’s activation induces the synthesis of pro-IL-1β and the activation of caspase 1 that induces the proteolytic maturation of pro-IL-1β [31]. Caspase 1 and ASC are key components of NLRP3 inflammasome, a multiproteic complex that induces inflammatory responses and cell death in response to various danger signals [66], including pro-inflammatory cytokines, reactive species, oxidized compounds that are known to accumulate in the muscle and blood during exhaustive physical exercise [67,68,69]. Therefore, the effect of caffeine on the inflammatory response induced by LPS that we observed in the mouse muscle suggests that part of its ergogenic effects might be mediated by the inhibition of the inflammasome assembly.

The anti-inflammatory effect might also be related to the positive effects we observed on Adoras’ increased expression in the inflamed muscle. Thus, increasing the antagonism of Adoras will potentiate the anti-inflammatory effect on different immune cell populations, that are cells known for their expression of high levels of Adora2A. Accordingly, it has been proposed that caffeine is an immunosuppressor since it has shown to inhibit proliferation, activation, and cytokine secretion by lymphocytes [70]. For example, it has been shown that caffeine reduced TNF-a secretion and enhanced the expression of Adora2A in LPS-activated human macrophages [70]. In addition, reduced ATP/AMP ratio, which occurs during the inflammatory response [71], is a key modulator of enhanced AMPK signaling, which denotes energy deficit. Although, we did not measure the levels of phosphorylated AMPK, the restoration of basal *Ampk* gene expression suggests that caffeine also protects the inflamed muscle by improving energy metabolism as previously proposed [72].

DNA methylation is also known to repress gene expression by blocking the promoter sites at which activating transcription factors are bound [73]. The reduced global DNA demethylation could be responsible for the upregulation of *Ampk* under LPS treatment, which was rescued when caffeine was also administered. This is also in agreement with the increased expression of pro-inflammatory cytokines, which was negatively modulated after the coadministration of caffeine in our study. This effect has also been reported in other tissues and cells. For example, LPS-challenged human peripheral blood mononuclear cells exposed to caffeine at different concentrations (10–100 μM) for 24 h, negatively modulated the production of TNF-α [74]. Similarly, mouse splenocyte cultures stimulated with concanavalin A (a pro-inflammatory agent) showed reduced production of TNF-α, IL-2, and IFN-γ when co-treated with 3.75 and 10 mM of caffeine for 24 h [75].

The DNA methylation machinery requires DNMT3a and DNMT3b for the de novo [76], and DNMT1 for the maintenance [77] of DNA methylation. In general, when methylation occurs in the promoter region of a particular gene, the gene expression is expected to be repressed. DNA can also be demethylated by the action of ten-eleven translocation (TET) enzymes TET1, TET2, and TET3 [78], which may result in enhanced gene expression. Therefore, the balance of these processes may regulate the expression of different genes, including the ones involved in inflammation and adenosine signaling as shown here. Indeed, genome-wide meta-analyses identified several genes positively associating caffeine consumption and DNA methylation [22,79,80]. While previous studies have shown that caffeine intake is positively correlated with higher DNA methylation [22], we have shown in this study that caffeine *per se* can be responsible for the negative modulation of the expression of inflammatory genes in animals submitted to acute inflammation.

## 5. Conclusions

This study provides evidence for the anti-inflammatory effect of caffeine in the mouse muscle. The immune system activated by LPS induced the release of pro-inflammatory cytokines that was prevented by caffeine administration, an effect also observed in the reduction of the inflammasome components, possibly by a modulatory effect of caffeine on epigenetics.

## Figures and Tables

**Figure 1 antioxidants-12-00554-f001:**
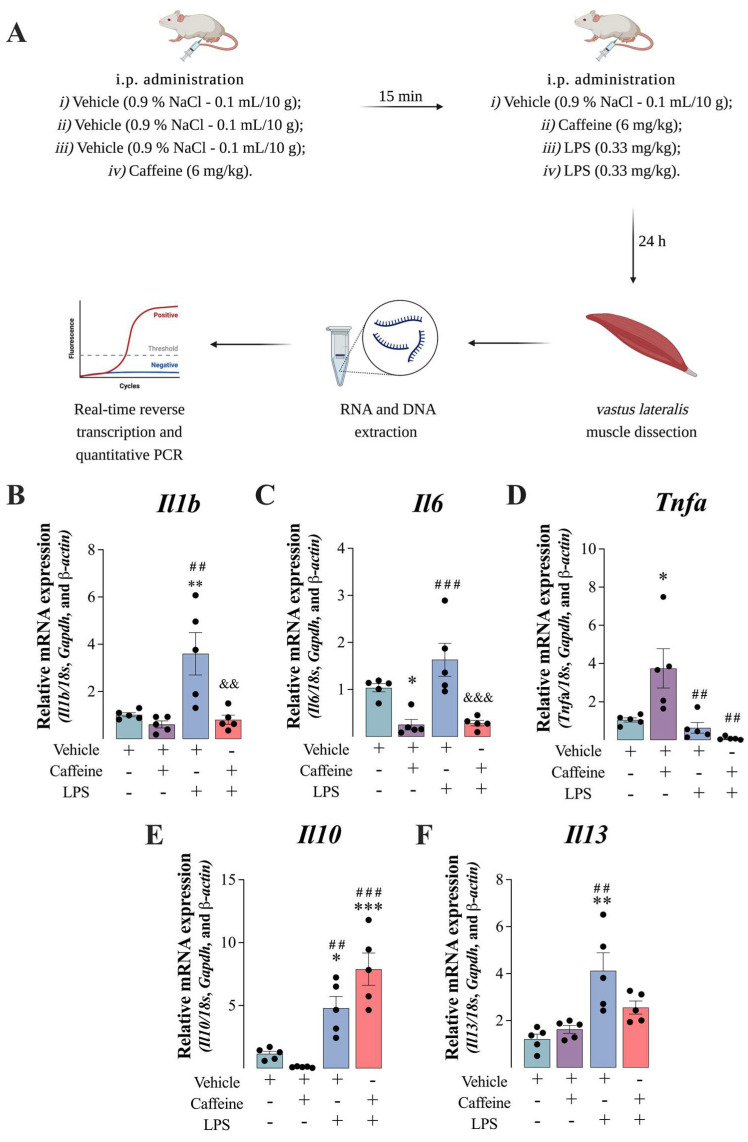
Caffeine administration reduced lipopolysaccharide (LPS)-mediated inflammation in the mouse *vastus lateralis* muscle. Adult Swiss male mice (3–5 months of age; body mass 45–50 g) received a single intraperitoneal (i.p.) injection of caffeine and/or LPS (See Materials and Methods for details). Schematic representation of the experimental protocol used to induced LPS-mediated inflammation in mouse, the muscle dissection and RNA/DNA extraction for the PCR analysis (**A**). The transcriptional profile of the cytokines *Il1b* (**B**), *Il*6 (**C**), *Tnfa* (**D**), *Il10* (**E**) and *Il13* (**F**) were evaluated by qPCR after the total RNA extraction (TRIzol^®^/Chloroform/Isopropanol method) from the muscle. Gene expression raw data were normalized by the average of the Ct of the *18s*, *Gapdh* and β-*actin* genes and calculated by the method (2^−ΔCt^). Bars represent the mean ± standard error of mean of five independent experiments (animals) performed in technical duplicates. * *p* < 0.05; ** *p* < 0.01; and *** *p* < 0.001 vs. vehicle; ^##^ *p* < 0.01; ^###^ *p* < 0.001 vs. to caffeine, and ^&&^ *p* < 0.01 and ^&&&^ *p* < 0.001 vs. LPS. Two-way ANOVA followed by Tukey’s test.

**Figure 2 antioxidants-12-00554-f002:**
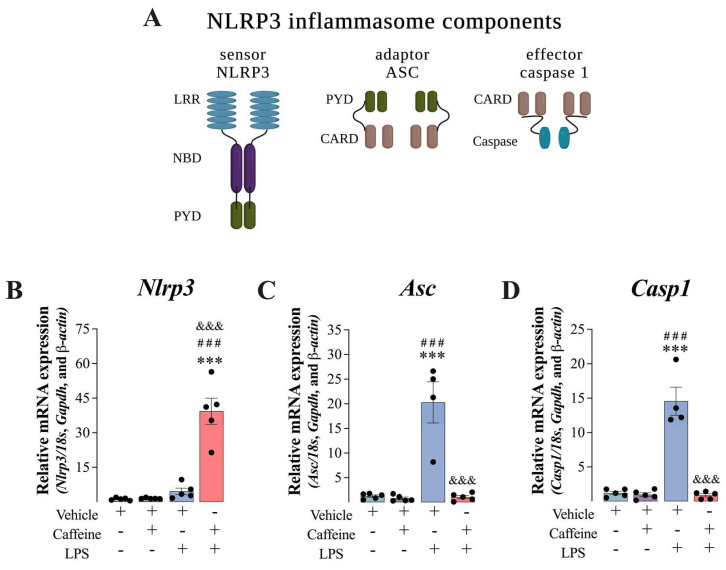
Caffeine administration rescued lipopolysaccharide (LPS)-induced Nrlp3 inflammasome components upregulation in the *vastus lateralis* muscle of mice. Adult Swiss male mice (3–5 months of age; body mass 45–50 g) received a single intraperitoneal (i.p.) injection of caffeine and/or LPS (See Materials and Methods for details). NLRP3 inflammasome consists of a sensor (Nlrp3), an adaptor (Asc) and an effector (Caspase-1) (**A**). The transcriptional profile of the components of the NLRP3 inflammasome *Nlrp3* (**B**), *Asc1* (**C**), and *Casp1* (**D**). Gene expression raw data were normalized by the average of the Ct of the *18s*, *Gapdh* and β-*actin* genes and calculated by the method (2^−ΔCt^). Bars represent the mean ± standard error of the mean of five independent experiments (animals) performed in technical duplicates. *** *p* < 0.001 vs. vehicle; ^###^ *p* < 0.001 vs. to caffeine, and ^&&&^ *p* < 0.001 vs. LPS. Two-way ANOVA followed by Tukey’s test.

**Figure 3 antioxidants-12-00554-f003:**
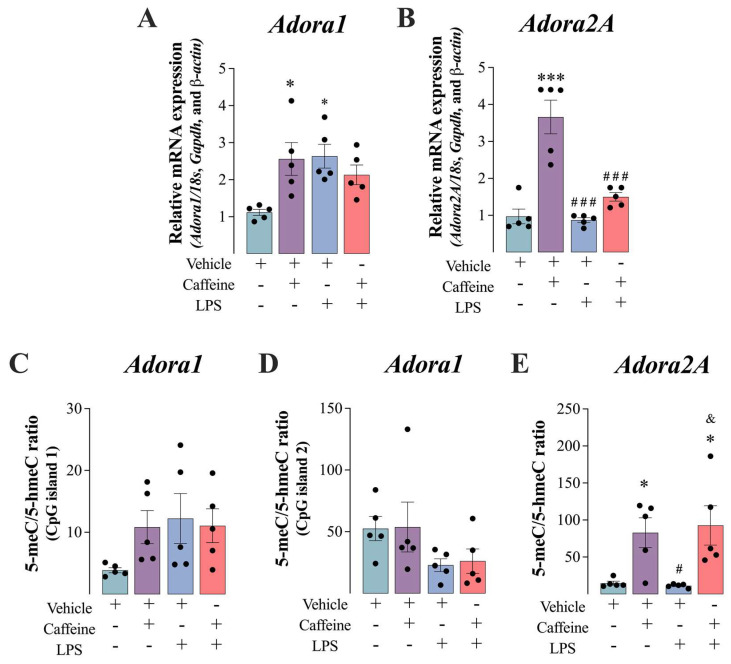
Caffeine administration induced the upregulation of genes encoding adenosinergic receptors in the *vastus lateralis* muscle of mice. Adult Swiss male mice (3–5 months of age; body mass 45–50 g) received a single intraperitoneal (i.p.) injection of caffeine and/or LPS (See Materials and Methods for details). The transcriptional profile of the components of the adenosine receptors *Adora1* (**A**), and *Adora2A* (**B**). DNA methylation and demethylation of *Adora1* CpG island 1 (**C**), *Adora1* Cpg island 2 (**D**), and *Adora2A* (**E**). Gene expression raw data were normalized by the average of the Ct of the *18s*, *Gapdh* and β-*actin* genes and calculated by the method (2^−ΔCt^). Bars represent the mean ± standard error of the mean of five independent experiments (animals) performed in technical duplicates. * *p* < 0.05; *** *p* < 0.001 vs. vehicle; ^#^ *p* < 0.05; ^###^ *p* < 0.001 vs. to caffeine, and ^&^ *p* < 0.05 vs. LPS. Two-way ANOVA followed by Tukey’s test.

**Figure 4 antioxidants-12-00554-f004:**
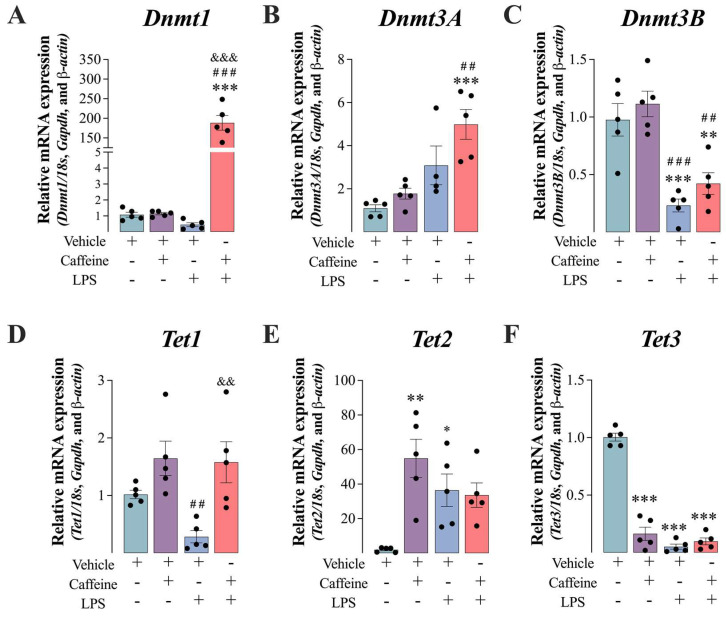
Caffeine plus lipopolysaccharide (LPS) administration induced de novo DNA methylation in the *vastus lateralis* muscle of mice. Adult Swiss male mice (3–5 months of age; body mass 45–50 g) received a single intraperitoneal (i.p.) injection of caffeine and/or LPS (See Materials and Methods for details). The transcriptional profile of the de novo DNA methylation of *Dnmt1* (**A**), *Dnmt2A* (**B**), and *Dnmt3B* (**C**). The status of the DNA demethylation enzymes *Tet1* (**D**), *Tet2* (**E**), and *Tet3* (**F**). Gene expression raw data were normalized by the average of the Ct of the *18s*, *Gapdh* and β-*actin* genes and calculated by the method (2^−ΔCt^). Bars represent the mean ± standard error of the mean of five independent experiments (animals) performed in technical duplicates. * *p* < 0.05; ** *p* < 0.01; *** *p* < 0.001 vs. vehicle; ^##^ *p* < 0.01; ^###^ *p* < 0.001 vs. caffeine and ^&&^ *p* < 0.01 and ^&&&^ *p* < 0.001 vs. LPS. Two-way ANOVA followed by Tukey’s test.

**Figure 5 antioxidants-12-00554-f005:**
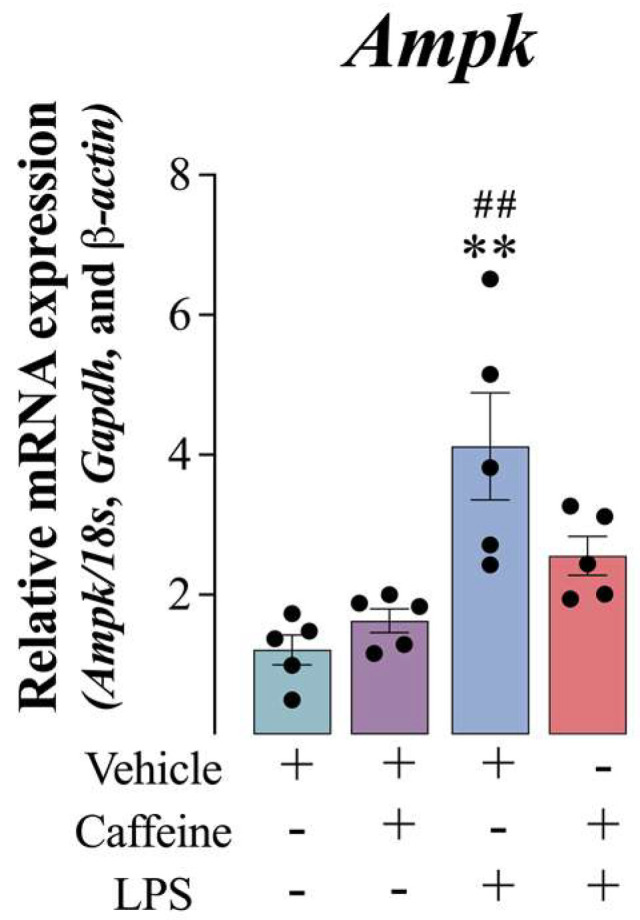
Caffeine administration reverted the catabolic effect induced by the administration of lipopolysaccharide (LPS) in the *vastus lateralis* muscle of mice. Adult Swiss male mice (3–5 months of age; body mass 45–50 g) received a single intraperitoneal (i.p.) injection of caffeine and/or LPS (See Materials and Methods for details). *Ampk* gene expression raw data were normalized by the average of the Ct of the *18s*, *Gapdh* and β-*actin* genes and calculated by the method (2^−ΔCt^). Bars represent the mean ± standard error of the mean of five independent experiments (animals) performed in technical duplicates. ** *p* < 0.01 vs. vehicle; ^##^ *p* < 0.01 vs. caffeine. Two-way ANOVA followed by Tukey’s test.

**Table 1 antioxidants-12-00554-t001:** Oligonucleotide primers and PCR conditions used in quantitative real-time PCR.

Gene (ID)	Primer	5′-3′Sequence	Reaction Condition	Product Size (bp)
*Il1b* (16176)	Forward	GAC CTT GGA TGA GGA CA	95 °C-15 s; 60 °C-30 s; 72 °C-30 s	183
	Reverse	AGC TCA TAT GGG TCC GAC AG		
*Il6* (16193)	Forward	AGT TCG CTT CTT GGG ACT GA	95 °C-15 s; 60 °C-30 s; 72 °C-30 s	191
	Reverse	CAG AAT TGC CAT TGC ACA AC		
*Tnfa* (11647)	Forward	CCA CAT CTC CCT CCA GAA AA	95 °C-15 s; 60 °C-30 s; 72 °C-30 s	259
	Reverse	AGG GTC TGG GCC ATA GAA CT		
*Il10* (21926)	Forward	CCA AGC CCT TAT CGG AAA TGA	95 °C-15 s; 60 °C-30 s; 72 °C-30 s	163
	Reverse	TTT TCA CAG GGG AGA AAT CG		
*Il13* (16163)	Forward	CAG TCC TGG CTC TTG CTT G	95 °C-15 s; 60 °C-30 s; 72 °C-30 s	165
	Reverse	CCA GGT CCA CAC TCC ATA CC		
*Adora1* (11539)	Forward	AGA ACC ACC TCC ACC CTT CT	95 °C-15 s; 63 °C-30 s; 72 °C-30 s	227
	Reverse	TAC TCT GGG TGG TGG TCA CA		
*Adora2A* (11540)	Forward	ATC CCT CAGAGA AGG GAA GC	95 °C-15 s; 63 °C-30 s; 72 °C-30 s	300
	Reverse	AGC TTC CCA AAG GCT TTC TC		
*Dnmt1* (13433)	Forward	CCT TTG TGG GAA CCT GGA A	95 °C-15 s; 63 °C-30 s; 72 °C-30 s	240
	Reverse	CTG TCG TCT GCG GTG ATT		
*Dnmt3A* (13435)	Forward	GAG GGA ACT GAG ACC CCA C	95 °C-15 s; 63 °C-30 s; 72 °C-30 s	216
	Reverse	CTG GAA GGT GAG TCT TGG CA		
*Dnmt3B* (113436)	Forward	AGC GGG TAT GAG GAG TGC AT	95 °C-15 s; 63 °C-30 s; 72 °C-30 s	91
	Reverse	GGG AGC ATC CTT CGT GTC TG		
*Tet1* (52463)	Forward	GAG CCT GTT CCT CGA TGT GG	95 °C-15 s; 65 °C-30 s; 72 °C-30 s	367
	Reverse	CAA ACC CAC CTG AGG CTG TT		
*Tet2* (214133)	Forward	AAC CTG GCT ACT GTC ATT GCT CCA	95 °C-15 s; 65 °C-30 s; 72 °C-30 s	211
	Reverse	ATG TTC TGC TGG TCT CTG TGG GAA		
*Tet3* (194388)	Forward	GTC TCC CCA AGT CCT ACC TCC G	95 °C-15 s; 63 °C-30 s; 72 °C-30 s	137
	Reverse	GTC AGT GCC CCA CGC TTC A		
*b-actin* (11461)	Forward	TCT TGG GTA TGG AAT CCT GTG	95 °C-15 s; 58 °C-30 s; 72 °C-30 s	82
	Reverse	AGG TCT TTA CGG ATG TCA ACG		
*Gapdh* (14433)	Forward	AGG CCG GTG CTG AGT ATG TC	95 °C-15 s; 58 °C-30 s; 72 °C-30 s	530
	Reverse	TGC CTG CTT CAC CAC CTT CT		

## Data Availability

The data presented in this study are available on request from the corresponding author.

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
