# Peer review of "Anti-Inflammatory Effect of Caffeine on Muscle under Lipopolysaccharide-Induced Inflammation"

_antioxidants, 2023, doi:10.3390/antiox12030554_

Round 1
Reviewer 1 Report
In the article titled: „Anti-inflammatory effect of caffeine on muscle under lipopolysaccharide-induced inflammation” Tuany Eichwald et al. examined the effects of caffeine administration on the expression of inflammatory, adenosine receptors, epigenetics, and oxidative metabolism-linked genes in the vastus lateralis muscle of mice submitted to LPS-induced inflammation. Manuscript needs improvement:
1. in the introduction there is no description of what is the content of the publication - one sentence at the end of the introduction is not informative.
2. please change the font in the figures - for example, fig 1 is illegible (IL-1, TNFalpha)
3. please complete the description of the collection of muscle tissue, the method of killing the animals and possible storage of the collected tissue.
4. Discussion should be enriched with information about the genes studied - lacking to justify the research conducted.
Author Response
RESPONSES TO THE REVIEWER #1
Comments and Suggestions for Authors
In the article titled: „Anti-inflammatory effect of caffeine on muscle under lipopolysaccharide-induced inflammation” Tuany Eichwald et al. examined the effects of caffeine administration on the expression of inflammatory, adenosine receptors, epigenetics, and oxidative metabolism-linked genes in the vastus lateralis muscle of mice submitted to LPS-induced inflammation.
Manuscript needs improvement:
- in the introduction there is no description of what is the content of the publication - one sentence at the end of the introduction is not informative.
Response:
We extensively modified the introduction section to be more informative, and to describe better our objectives and hypothesis. As a consequence, we also added key references that were missing.
- Introduction
Regular moderate intensity exercise is proven to promote an anti-inflammatory state that helps to prevent the development of chronic diseases [for a review see (Scheffer and Latini, 2020)]. Strenuous exercise can lead to increased levels of blood proinflammatory cytokines, which are linked to fatigue, and therefore, to reduced performance (Hargreaves and Spriet, 2020). This scenario has seen caffeine administration used to increase alertness (Vital-Lopez et al., 2018), to accelerate metabolism (Harpaz et al., 2017), and to delay fatigue development in aerobic and anaerobic exercises, including muscular strength (Astorino et al., 2007), running (Graham and Spriet, 1991), cycling (Spriet et al., 1992), team sports (Schneiker et al., 2006), among others.
While it is unclear what molecular mechanisms are behind caffeine consumption and its ergogenic responses, evidence is mounting that caffeine may induce anti-inflammatory effects in both humans and animals. For example, it was demonstrated that caffeine supplementation reduced proinflammatory markers in the blood of athletes (Rodas et al., 2020; Tauler et al., 2016, 2013). In the case of animal models, reduced proinflammatory and increased anti-inflammatory markers were not only seen in the blood of trained rats, but also in key tissues linked to exercise performance, like the brain, the lung, the heart and the skeletal muscle of rodents exposed to caffeine (Barcelos et al., 2014; Endesfelder et al., 2020; Farokhi-Sisakht et al., 2022; Jia et al., 2014; Yang et al., 2022). Furthermore, an elegant study involving 114 participants showed that caffeine intake is associated with lower inflammation and activation of the inflammasome, which resulted in less production of the proinflammatory cytokyne interleukin-1 beta (IL-1ß) (Furman et al., 2017).
In addition, caffeine supplementation has also been shown to cause changes in gene expression that could be linked to improved exercise performance (Egan et al., 2013; Egan and Zierath, 2013; Seaborne et al., 2018). These modifications have been related to altered epigenetics, a term conceived to describe the possible causal processes acting on genes that regulate phenotype (Waddington, 2012). Some of the reported effects of caffeine are associated with DNA methylation, a major epigenetic factor influencing gene activities (Chuang et al., 2017). Considering that epigenetics can change the activity of a DNA segment without changing the sequence, it is plausible that caffeine can modulate inflammatory processes by changing the epigenetic landscape. When DNA methylation is increased in a gene promoter, it will typically act to repress gene transcription, including the expression of inflammatory mediators. Altogether, we aimed to understand whether caffeine can modulate epigenetics to induce an anti-inflammatory scenario in the mouse skeletal muscle.
- please change the font in the figures - for example, fig 1 is illegible (IL-1, TNFalpha)
Response:
As suggested, the font of the figures was modified to make the content of the graphs clearer (all figures were included in the revised version of the manuscript and also in the last page of this file).
- please complete the description of the collection of muscle tissue, the method of killing the animals and possible storage of the collected tissue.
Response:
Following the suggestion of the reviewer, we added the following in the Material and Methods section (page= 2; line= 83).
Twenty-four h after the treatments mice were euthanized by cervical dislocation and the vastus lateralis muscle was immediately collected and processed in Trizol as previously published by our group (remor et al 2018). The dosage of LPS used was based on previously published data (de Paula Martins et al., 2018; Ghisoni et al., 2015).
- Discussion should be enriched with information about the genes studied - lacking to justify the research conducted.
Response:
To address the issue risen by the reviewer, we added the following in the Discussion section (page= 11, line= 397)
The DNA methylation machinery requires DNMT3a and DNMT3b for the de novo (Okano et al., 1998), and DNMT1 for the maintenance (Hermann et al., 2004) of DNA methylation. In general, when methylation occurs in the promoter region of a particular gene, the gene expression is expected to be repressed. DNA can also be demethylated by the action of ten-eleven translocation (TET) enzymes TET1, TET2, and TET3 (Ito et al., 2010), which may result in enhanced gene expression. Therefore, the balance of these processes may regulate the expression of different genes, including the ones involved in inflammation and adenosine signaling as shown here. Indeed, a genome-wide meta-analysis identified several genes positively associating caffeine consumption and DNA methylation (Chuang et al., 2017; Cornelis et al., 2015, 2011). While previous studies have shown that caffeine intake positively correlated with higher DNA methylation (Chuang et al., 2017), we have shown in this study that caffeine per se can be responsible for the negative modulation of the expression of inflammatory genes in animals submitted to acute inflammation.
Finally, we would like to add that the text was thoroughly revised and edited by an English native speaker.
References cites in responses to points 1 and 4
Astorino TA, Rohmann RL, Firth K. 2007. Effect of caffeine ingestion on one-repetition maximum muscular strength. Eur J Appl Physiol 102:127–132. doi:10.1007/s00421-007-0557-x
Barcelos R, Souza M, Amaral G, Stefanello S, Bresciani G, Fighera M, Soares F, de Vargas Barbosa N. 2014. Caffeine Intake May Modulate Inflammation Markers in Trained Rats. Nutrients 6:1678–1690. doi:10.3390/nu6041678
Chuang Y-H, Quach A, Absher D, Assimes T, Horvath S, Ritz B. 2017. Coffee consumption is associated with DNA methylation levels of human blood. Eur J Hum Genet 25:608–616. doi:10.1038/ejhg.2016.175
Cornelis MC, Byrne EM, Esko T, Nalls MA, Ganna A, Paynter N, Monda KL, Amin N, Fischer K, Renstrom F, Ngwa JS, Huikari V, Cavadino A, Nolte IM, Teumer A, Yu K, Marques-Vidal P, Rawal R, Manichaikul A, Wojczynski MK, Vink JM, Zhao JH, Burlutsky G, Lahti J, Mikkilä V, Lemaitre RN, Eriksson J, Musani SK, Tanaka T, Geller F, Luan J, Hui J, Mägi R, Dimitriou M, Garcia ME, Ho W-K, Wright MJ, Rose LM, Magnusson PKE, Pedersen NL, Couper D, Oostra BA, Hofman A, Ikram MA, Tiemeier HW, Uitterlinden AG, van Rooij FJA, Barroso I, Johansson I, Xue L, Kaakinen M, Milani L, Power C, Snieder H, Stolk RP, Baumeister SE, Biffar R, Gu F, Bastardot F, Kutalik Z, Jacobs DR, Forouhi NG, Mihailov E, Lind L, Lindgren C, Michaëlsson K, Morris A, Jensen M, Khaw K-T, Luben RN, Wang JJ, Männistö S, Perälä M-M, Kähönen M, Lehtimäki T, Viikari J, Mozaffarian D, Mukamal K, Psaty BM, Döring A, Heath AC, Montgomery GW, Dahmen N, Carithers T, Tucker KL, Ferrucci L, Boyd HA, Melbye M, Treur JL, Mellström D, Hottenga JJ, Prokopenko I, Tönjes A, Deloukas P, Kanoni S, Lorentzon M, Houston DK, Liu Y, Danesh J, Rasheed A, Mason MA, Zonderman AB, Franke L, Kristal BS, Karjalainen J, Reed DR, Westra H-J, Evans MK, Saleheen D, Harris TB, Dedoussis G, Curhan G, Stumvoll M, Beilby J, Pasquale LR, Feenstra B, Bandinelli S, Ordovas JM, Chan AT, Peters U, Ohlsson C, Gieger C, Martin NG, Waldenberger M, Siscovick DS, Raitakari O, Eriksson JG, Mitchell P, Hunter DJ, Kraft P, Rimm EB, Boomsma DI, Borecki IB, Loos RJF, Wareham NJ, Vollenweider P, Caporaso N, Grabe HJ, Neuhouser ML, Wolffenbuttel BHR, Hu FB, Hyppönen E, Järvelin M-R, Cupples LA, Franks PW, Ridker PM, van Duijn CM, Heiss G, Metspalu A, North KE, Ingelsson E, Nettleton JA, van Dam RM, Chasman DI. 2015. Genome-wide meta-analysis identifies six novel loci associated with habitual coffee consumption. Mol Psychiatry 20:647–656. doi:10.1038/mp.2014.107
Cornelis MC, Monda KL, Yu K, Paynter N, Azzato EM, Bennett SN, Berndt SI, Boerwinkle E, Chanock S, Chatterjee N, Couper D, Curhan G, Heiss G, Hu FB, Hunter DJ, Jacobs K, Jensen MK, Kraft P, Landi MT, Nettleton JA, Purdue MP, Rajaraman P, Rimm EB, Rose LM, Rothman N, Silverman D, Stolzenberg-Solomon R, Subar A, Yeager M, Chasman DI, van Dam RM, Caporaso NE. 2011. Genome-Wide Meta-Analysis Identifies Regions on 7p21 (AHR) and 15q24 (CYP1A2) As Determinants of Habitual Caffeine Consumption. PLoS Genet 7:e1002033. doi:10.1371/journal.pgen.1002033
de Paula Martins R, Glaser V, Aguiar AS, de Paula Ferreira PM, Ghisoni K, da Luz Scheffer D, Lanfumey L, Raisman-Vozari R, Corti O, De Paul AL, da Silva RA, Latini A. 2018. De novo tetrahydrobiopterin biosynthesis is impaired in the inflammed striatum of parkin(-/-) mice. Cell Biol Int 42:725–733. doi:10.1002/cbin.10969
Egan B, O’Connor PL, Zierath JR, O’Gorman DJ. 2013. Time Course Analysis Reveals Gene-Specific Transcript and Protein Kinetics of Adaptation to Short-Term Aerobic Exercise Training in Human Skeletal Muscle. PLoS One. doi:10.1371/journal.pone.0074098
Egan B, Zierath JR. 2013. Exercise metabolism and the molecular regulation of skeletal muscle adaptation. Cell Metab. doi:10.1016/j.cmet.2012.12.012
Endesfelder S, Strauß E, Bendix I, Schmitz T, Bührer C. 2020. Prevention of Oxygen-Induced Inflammatory Lung Injury by Caffeine in Neonatal Rats. Oxid Med Cell Longev. doi:10.1155/2020/3840124
Farokhi-Sisakht F, Farhoudi M, Mahmoudi J, Farajdokht F, Kahfi-Ghaneh R, Sadigh-Eteghad S. 2022. Effect of intranasal administration of caffeine on mPFC ischemia‑induced cognitive impairment in BALB/c mice. Acta Neurobiol Exp82:295–303. doi:10.55782/ane-2022-028
Furman D, Chang J, Lartigue L, Bolen CR, Haddad F, Gaudilliere B, Ganio EA, Fragiadakis GK, Spitzer MH, Douchet I, Daburon S, Moreau JF, Nolan GP, Blanco P, Déchanet-Merville J, Dekker CL, Jojic V, Kuo CJ, Davis MM, Faustin B. 2017. Expression of specific inflammasome gene modules stratifies older individuals into two extreme clinical and immunological states. Nat Med. doi:10.1038/nm.4267
Ghisoni K, Martins RDPR de P, Barbeito L, Latini A. 2015. Neopterin as a potential cytoprotective brain molecule. J Psychiatr Res 71:134–139. doi:10.1016/j.jpsychires.2015.10.003
Graham TE, Spriet LL. 1991. Performance and metabolic responses to a high caffeine dose during prolonged exercise. J Appl Physiol 71:2292–2298. doi:10.1152/jappl.1991.71.6.2292
Hargreaves M, Spriet LL. 2020. Skeletal muscle energy metabolism during exercise. Nat Metab 2:817–828. doi:10.1038/s42255-020-0251-4
Harpaz E, Tamir S, Weinstein A, Weinstein Y. 2017. The effect of caffeine on energy balance. J Basic Clin Physiol Pharmacol. doi:10.1515/jbcpp-2016-0090
Hermann A, Goyal R, Jeltsch A. 2004. The Dnmt1 DNA-(cytosine-C5)-methyltransferase methylates DNA processively with high preference for hemimethylated target sites. J Biol Chem 279:48350–48359. doi:10.1074/jbc.M403427200
Ito S, D’Alessio AC, Taranova O V., Hong K, Sowers LC, Zhang Y. 2010. Role of Tet proteins in 5mC to 5hmC conversion, ES-cell self-renewal and inner cell mass specification. Nature 466:1129–1133. doi:10.1038/nature09303
Jia H, Aw W, Egashira K, Takahashi S, Aoyama S, Saito K, Kishimoto Y, Kato H. 2014. Coffee intake mitigated inflammation and obesity-induced insulin resistance in skeletal muscle of high-fat diet-induced obese mice. Genes Nutr. doi:10.1007/s12263-014-0389-3
Okano M, Xie S, Li E. 1998. Cloning and characterization of a family of novel mammalian DNA (cytosine-5) methyltransferases. Nat Genet 19:219–220. doi:10.1038/890
Rodas L, Martinez S, Aguilo A, Tauler P. 2020. Caffeine supplementation induces higher IL-6 and IL-10 plasma levels in response to a treadmill exercise test. J Int Soc Sports Nutr. doi:10.1186/s12970-020-00375-4
Scheffer D da L, Latini A. 2020. Exercise-induced immune system response: Anti-inflammatory status on peripheral and central organs, Biochimica et Biophysica Acta - Molecular Basis of Disease. doi:10.1016/j.bbadis.2020.165823
Schneiker KT, Bishop D, Dawson B, Hackett LP. 2006. Effects of Caffeine on Prolonged Intermittent-Sprint Ability in Team-Sport Athletes. Med Sci Sport Exerc 38:578–585. doi:10.1249/01.mss.0000188449.18968.62
Seaborne RA, Strauss J, Cocks M, Shepherd S, O’Brien TD, Van Someren KA, Bell PG, Murgatroyd C, Morton JP, Stewart CE, Sharples AP. 2018. Human Skeletal Muscle Possesses an Epigenetic Memory of Hypertrophy. Sci Rep. doi:10.1038/s41598-018-20287-3
Spriet LL, MacLean DA, Dyck DJ, Hultman E, Cederblad G, Graham TE. 1992. Caffeine ingestion and muscle metabolism during prolonged exercise in humans. Am J Physiol Metab 262:E891–E898. doi:10.1152/ajpendo.1992.262.6.E891
Tauler P, Martinez S, Martinez P, Lozano L, Moreno C, Aguiló A. 2016. Effects of caffeine supplementation on plasma and blood mononuclear cell interleukin-10 levels after exercise. Int J Sport Nutr Exerc Metab 26:8–16. doi:10.1123/ijsnem.2015-0052
Tauler P, Martínez S, Moreno C, Monjo M, Martínez P, Aguiló A. 2013. Effects of caffeine on the inflammatory response induced by a 15-km run competition. Med Sci Sports Exerc 45:1269–1276. doi:10.1249/MSS.0b013e3182857c8a
Vital-Lopez FG, Ramakrishnan S, Doty TJ, Balkin TJ, Reifman J. 2018. Caffeine dosing strategies to optimize alertness during sleep loss. J Sleep Res. doi:10.1111/jsr.12711
Waddington CH. 2012. The epigenotype. 1942. Int J Epidemiol 41:10–13. doi:10.1093/ije/dyr184
Yang L, Yu X, Zhang Y, Liu N, Xue X, Fu J. 2022. Caffeine treatment started before injury reduces hypoxic–ischemic white-matter damage in neonatal rats by regulating phenotypic microglia polarization. Pediatr Res. doi:10.1038/s41390-021-01924-6

Reviewer 2 Report
This manuscript describes a study showing that epigenetics is involved in the anti-inflammatory effects of caffeine on the vastus lateralis muscle of resting mice. The results show that the treatment with caffeine prevented the increase the gene expression of LPS-induced pro-inflammatory cytokines Il1beta and Il6 and promoted the upregulation of the anti-inflammatory genes Il10 and Il13 in the mouse muscle. The experiments were carried out properly and the results were presented clearly. Beneficial aspects of caffeine have been reported in many articles so far and this report has focused on ani-inflammatory characteristics of caffeine, which is interesting. The authors should pay attention to the points below.
Issues to be considered
1 Inflammatory what? (line 6)
2 Spin should read as spun (line 116)
3 The following sentence seems to make no sense to the context and should be reconsidered.
”This section may be divided by subheadings. It should provide a concise and precise description of the experimental results, their interpretation, as well as the experimental conclusions that can be drawn.(line 277-278)
”
4 Please explain the altered expression pattern of NLRP3 compared with those of an adaptor (Asc) and an effector (Caspase-1) (Fig. 2B). As stated in the text, although, LPS treatment per se did not alter the levels of Nlrp3 expression 24 h after the administration, the combination with caffeine provoked its upregulation. Why is it?
Author Response
RESPONSES TO THE REVIEWER #2
Comments and Suggestions for Authors
This manuscript describes a study showing that epigenetics is involved in the anti-inflammatory effects of caffeine on the vastus lateralis muscle of resting mice. The results show that the treatment with caffeine prevented the increase the gene expression of LPS-induced pro-inflammatory cytokines Il1beta and Il6 and promoted the upregulation of the anti-inflammatory genes Il10 and Il13 in the mouse muscle. The experiments were carried out properly and the results were presented clearly. Beneficial aspects of caffeine have been reported in many articles so far and this report has focused on ani-inflammatory characteristics of caffeine, which is interesting. The authors should pay attention to the points below.
Issues to be considered
1 Inflammatory what? (line 6)
Response:
We apologize to the reviewer that we could not find the issue related to inflammation in line 6. However, we noticed that the word inflammatory should have been preceded by the prefix “pro” in page=4, line= 164.
The sentence “Figure 1 shows the effect of caffeine and/or LPS administration (i.p.) after twenty-four h on inflammatory cytokines gene expression in the mouse vastus lateralis muscle (Fig. 1A).” now reads “Figure 1 shows the effect of caffeine and/or LPS administration (i.p.) after twenty-four h on pro-inflammatory cytokines gene expression in the mouse vastus lateralis muscle (Fig. 1A).”
2 Spin should read as spun (line 116)
Response:
The sentence was edited as indicated.
3 The following sentence seems to make no sense to the context and should be reconsidered.
”This section may be divided by subheadings. It should provide a concise and precise description of the experimental results, their interpretation, as well as the experimental conclusions that can be drawn.(line 277-278)
Response:
We deleted the sentence mentioned by the reviewer (lines= 277-278). The sentence was included in the template of the journal, so it was added by mistake.
4 Please explain the altered expression pattern of NLRP3 compared with those of an adaptor (Asc) and an effector (Caspase-1) (Fig. 2B). As stated in the text, although, LPS treatment per se did not alter the levels of Nlrp3 expression 24 h after the administration, the combination with caffeine provoked its upregulation. Why is it?
Response:
The coadministration of caffeine and LPS provoked a marked reduction of Asc and Caspase-1 gene expression in the mouse muscle, pointing to an anti-inflammatory effect of caffeine. This down-regulation might have been compensated by increasing the expression of the protein that works as a receptor of the inflammasome complex. Since this is the first time the effect of caffeine under LPS treatment has been reported in the mouse muscle, we are aware that more studies are needed to better understand how caffeine induces anti-inflammatory responses.
Finally, we would like to add that the text was thoroughly revised and edited by an English native speaker.

Round 2
Reviewer 1 Report
The manuscript has been significantly revised, and is suitable for publication in its present form.